# Accumulation of Experience and Newly Developed Devices Can Improve the Safety and Voice Outcome of Total Thyroidectomy for Graves’ Disease

**DOI:** 10.3390/jcm11051298

**Published:** 2022-02-27

**Authors:** Cheng-Hsun Chuang, Tzu-Yen Huang, Tzer-Zen Hwang, Che-Wei Wu, I-Cheng Lu, Pi-Ying Chang, Yi-Chu Lin, Ling-Feng Wang, Chih-Chun Wang, Ching-Feng Lien, Gianlorenzo Dionigi, Chih-Feng Tai, Feng-Yu Chiang

**Affiliations:** 1Department of Otorhinolaryngology-Head and Neck Surgery, Kaohsiung Medical University Hospital, Faculty of Medicine, College of Medicine, Kaohsiung Medical University, No. 100 Tzyou First Road, Kaohsiung 807, Taiwan; 18chengxun@gmail.com (C.-H.C.); tyhuang.ent@gmail.com (T.-Y.H.); cwwu@kmu.edu.tw (C.-W.W.); reddust0113@yahoo.com.tw (Y.-C.L.); lifewang@kmu.edu.tw (L.-F.W.); 2Department of Otolaryngology-Head and Neck Surgery, E-Da Hospital, No. 1, Yida Road, Jiaosu Village, Yanchao District, Kaohsiung 824, Taiwan; tzhwang@hotmail.com.tw (T.-Z.H.); ccw5969@yahoo.com.tw (C.-C.W.); lien980206@yahoo.com.tw (C.-F.L.); 3School of Medicine, College of Medicine, I-Shou University, Kaohsiung 824, Taiwan; 4Department of Anesthesiology, Kaohsiung Municipal Siaogang Hospital, Kaohsiung Medical University Hospital, Faculty of Medicine, College of Medicine, Kaohsiung Medical University, Kaohsiung 807, Taiwan; u9251112@gmail.com; 5Department of Anesthesiology, Kaohsiung Municipal Tatung Hospital, Kaohsiung Medical University Hospital, Faculty of Medicine, College of Medicine, Kaohsiung Medical University, Kaohsiung 801, Taiwan; annabelle69@gmail.com; 6Division of General Surgery, Endocrine Surgery Section, Istituto Auxologico Italiano IRCCS, 20095 Milan, Italy; gianlorenzo.dionigi@unimi.it; 7Department of Pathophysiology and Transplantation, University of Milan, 20133 Milan, Italy

**Keywords:** Graves’ disease, total thyroidectomy, major complications, voice outcome, experience and newly developed devices, energy-based device (EBD)

## Abstract

Total thyroidectomy (TT) in patients with Graves’ disease is challenging even for an experienced thyroid surgeon. This study aimed to investigate the accumulation of experience and applying newly developed devices on major complications and voice outcomes after surgery of a single surgeon over 30 years. This study retrospectively reviewed 90 patients with Graves’ disease who received TT. Forty-six patients received surgery during 1990–1999 (Group A), and 44 patients received surgery during 2010–2019 (Group B). Major complications rates were compared between Group A/B, and objective voice parameters were compared between the usage of energy-based devices (EBDs) within Group B. Compared to Group B, Group A patients had higher rates of recurrent laryngeal nerve palsy (13.0%/1.1%, *p* = 0.001), postoperative hypocalcemia (47.8%/18.2%, *p* = 0.002), and postoperative hematoma (10.9%/2.3%, *p* = 0.108). Additionally, Group A had one permanent vocal cord palsy, four permanent hypocalcemia, and one thyroid storm, whereas none of Group B had these complications. Group B patients with EBDs had a significantly better pitch range (*p* = 0.015) and jitter (*p* = 0.035) than those without EBDs. To reduce the major complications rate, inexperienced thyroid surgeons should remain vigilant when performing TT for Graves’ disease. Updates on surgical concepts and the effective use of operative adjuncts are necessary to improve patient safety and voice outcome.

## 1. Introduction

Graves’ disease is the most common cause of persistent hyperthyroidism. The annual prevalence rate is approximately 20 to 30 cases per 100,000 individuals, and the lifetime prevalence rates in women and men are 3% and 0.5%, respectively [1]. Depending on the preferences and clinical features of the patient, treatment for Graves’ disease may include anti-thyroid drugs, radioactive iodine therapy, and thyroidectomy [2].

The suggested indications for surgical treatment of Graves’ disease include large goiters, lesions causing tracheal compression, moderate-to-severe ophthalmopathy, current pregnancy or breastfeeding, poor control of hyperthyroidism after radio-iodine ablation or after anti-thyroid drug therapy, and suspected malignancy of a coexisting nodule [3]. In the literature, two surgical managements for Graves’ disease can be considered—subtotal or total thyroidectomy (TT). Subtotal thyroidectomy leaves 4 to 7 g of thyroid and provides patients with adequate thyroid function without requiring thyroxin replacement. Another advantage is that the procedure reduces the risk of hypoparathyroidism [4]. However, in a study of 415 consecutive Graves’ disease patients treated by subtotal thyroidectomy with a mean thyroid remnant weight of 5.1 g, 28.7% (119 patients) had persistent or recurrent hyperthyroidism, over 50% had hypothyroidism, and 19.3% achieved an euthyroid state [5]. Furthermore, a second surgery for recurrence may be more difficult than a first surgery owing to distortion of tissue planes by scar tissue formation and may have a higher risk of injury to the recurrent laryngeal nerve (RLN) and parathyroid glands (PGs) [6]. Therefore, TT is a preferred surgical treatment option because of several advantages, including (1) low recurrence rate, (2) rapid and reliable control of hyperthyroidism and its related symptoms, (3) radical resection of coexisting malignant thyroid tumor, (4) optimal release of airway compression, (5) absence of side-effects such as those in radio-iodine and anti-thyroid drug therapy, and (6) possible elimination of further progression of ophthalmopathy [7,8,9].

Compared to patients with euthyroid multiple nodular goiters, TT for Graves’ disease patients has significantly higher rates of vocal cord palsy, postoperative hypocalcemia (PH), and hematoma requiring reoperation [10]. Graves’ disease patients tend to have a larger thyroid size and an adhesive thyroid capsule to the surrounding neck structure, Palestini et al. [11] reported that voice changes or neck discomfort were reported by 29% and 8% of patients after thyroidectomy for Graves’ disease patients. Therefore, high levels of surgical experience and technical knowledge may have important roles in lowering the occurrence of the major complications and preventing voice impairment. To the best of our knowledge, this study is the first to investigate the surgical performance on major complications and voice outcome after TT for Graves’ disease of a single surgeon over 30 years.

## 2. Materials and Methods

This retrospective study enrolled patients with Graves’ disease who had received TT performed by a single thyroid surgeon (F.-Y.C.) within a 30 year period (1990–2019) in Kaohsiung Medical University Hospital. The surgeon performs more than 200 thyroid surgeries every year, and the surgical complications in recent years are presented in [12,13]. Ethical approval of this study was obtained from the Kaohsiung Medical University Hospital Institutional Review Board (KMUHIRB-E(II)-20200026). The surgeries were divided into two groups according to the time of surgery: Group A included 46 consecutive patients who had received TT during 1990–1999, and Group B included 44 consecutive patients who had received TT during 2010–2019. Thyroid surgeries with or without energy-based device (EBD) assistance in Group B were evaluated; in this study, the Ligasure^TM^ small jaw (Medtronic, Covidien, CO, USA) was applied as the EBD in Graves’ disease surgery.

The surgical indications for patients with Graves’ disease included refractory hyperthyroidism, suspected thyroid malignancy, local compression symptoms caused by Graves’ disease, and Graves’ ophthalmopathy. A different surgical strategy for preserving RLNs and PGs was used in each group. In Group A, all RLNs were identified and preserved by visualization alone, and preferably at least one PG was autotransplanted. In Group B, RLNs were routinely identified and preserved by visualization with the adjunct of intermittent intraoperative neuromonitoring (IONM), and PGs were preserved in situ whenever possible. Autotransplantation was the least preferred option for devascularized PGs.

In both groups of patients, vocal cord mobility was video-recorded with a flexible laryngo-fiberscope before and after surgery. If vocal cord palsy occurred, the patient was followed up until complete recovery of vocal cord function. An RLN palsy was considered permanent if vocal cord dysfunction persisted longer than 6 months after surgery. The RLN palsy rate was based on the number of nerves at risk.

Preoperative and postoperative (12, 24, 48, and 72 h after surgery) serum ionized calcium (iCa) levels were measured in each group. Normal iCa was defined as the mean (± 2SD) preoperative iCa. In Group A, PH was defined as iCa under 4.0 mg/dL in at least two measurements. In Group B, PH was defined as iCa under 4.2 mg/dL in at least two measurements. Permanent PH was defined as a persistent PH that required treatment with calcium supplements more than 12 months after surgery [13]. Postoperative hematoma was defined as progressive neck swelling that required emergent surgical intervention.

Objective voice analysis included the Multidimensional Voice Program (model 5105, version 3.1.7; KayPENTAX, NJ, USA) and Voice Range Profile (model 4326, version 3.3.0; KayPENTAX, NJ, USA). Objective voice parameters were obtained, including maximum pitch frequency (Fmax), minimum pitch frequency (Fmin), pitch range (PR), mean fundamental frequency (Mean F0), jitter, shimmer, and noise-to-harmonic ratio (NHR). The PR was defined as the number of semitones between Fmax and Fmin.

All patients received anti-thyroid drugs and were controlled to euthyroid state before surgery, and no patients in the two groups were given Lugol’s solution preoperatively.

Variables were analyzed by the t-test and chi-square test performed using SPSS (version 18.0 for windows; SPSS Inc., Chicago, IL, USA). A two-tailed *p* value less than 0.05 was considered statistically significant.

## 3. Results

Table 1 shows the demographic and clinical characteristics of the patients. There was no significant difference in gender, age, or pathology results between Group A and Group B. Among the patients in Group B, 3 of 7 patients having a malignant pathology result received radioactive iodine treatment, and none of the 7 patients received re-intervention or had cancer recurrence. However, the two groups significantly differed in the number of patients with at least one PG autotransplantation; forty (87.0%) patients in Group A and twenty-two (50.0%) patients in Group B had at least one PG autotransplantation (*p* = 0.001).

Table 2 compares the major complication rates in the two groups. The RLN palsy rate was significantly higher in Group A compared to Group B (13.0% vs. 1.1%, *p* = 0.001). Group A had 11 temporary palsies and one permanent palsy while Group B had only one temporary RLN palsy. Group A also had a significantly higher PH rate compared to Group B (47.8% vs. 18.2%, *p* = 0.002). Additionally, Group A had 18 (39.1%) patients with temporary PH and 4 (8.7%) patients with permanent PH, whereas Group B only had 8 (18.2%) patients with temporary PH and no patients with permanent PH.

Five (10.9%) patients in Group A and one (2.3%) patient in Group B developed postoperative hematoma, which showed no significant difference (*p* = 0.108). Only one patient in Group A had a thyroid storm.

Table 3 compares the changes in objective voice parameters between preoperative and 6 week postoperative periods in Group B with (n = 22) or without (n = 22) EBD assistance. The patients that received surgery with EBDs had a significantly lower proportion of PR decrease >30% (40.9% vs. 9.1%, *p* = 0.015) and Jitter increase >30% (63.6% vs. 31.8%, *p* = 0.035) compared to the without-EBD group.

## 4. Discussion

In comparison with Group A, Group B showed less incidence of major complications, particularly RLN palsy and PH. In Group B, surgery with EBD assistance showed better voice outcomes. Surgical experience, updates on surgical concepts, and effective use of operative adjuncts show a strong association to the surgical performance after TT in patients with Graves’ disease.

Duclos et al. [14] reported a permanent RLN palsy rate of 2.08% in 2357 patients under thyroid procedures, and the patients’ thyroid disease consisted of 69.8% (n = 1645) non-toxic nodule, 10.7% (n = 253) hyperthyroidism, 9.7% (n = 228) Grave’s disease, and 9.8% (n = 231) malignant neoplasm. In the group of Graves’ patients, the permanent RLN palsy rate was 3.5%. They demonstrated that the successful RLN preservation may be challenging even for experienced, high-volume surgeons. Wagner and Seiler also reported that, after thyroidectomy, patients with Graves’ disease had a higher rate of permanent RLN palsy compared to those with euthyroid nodular goiter (4% vs. 1.7%, respectively) [15]. All Group A patients in this study had received surgery performed by a single surgeon in his first decade of clinical practice. The RLN palsy rate reached as high as 13.0% when the surgeon had a relatively low experience level and did not use innovative surgical techniques such as IONM. Although the RLN was visually identified in all cases in Group A, 11 temporary palsies and one permanent palsy occurred. In other words, the visual integrity of the RLN is not consistent with the functional preservation, especially the temporary nerve injuries. In contrast, the RLN palsy rate decreased to 1.1% in Group B patients who received the surgeon’s third decade of practice and with the aid of IONM. The comparison suggests that the accumulated experience of the surgeon and the assistance of IONM provided the accuracy in identifying RLN, understood the mechanism of nerve injury, and improved the surgical technique, which resulted in a reduced RLN palsy rate in Group B.

Patients with thyroiditis do not have abnormal perceptual vocal evaluation or acoustic findings compared with controls [16]; however, the mass effect of Graves’ disease can be a negative factor for patients’ voice. Voice changes or neck discomfort were not uncommon after thyroidectomy for Graves’ disease patients [11]. For better surgical field exposure, Ko et al. [17] reported U-shape muscle flap (USMF) surgical methods and included two patients with large Graves’ disease in that study. They concluded that voice and swallowing functions after USMF are comparable to those obtained by the midline approach. Liu et al. [12] reported a 1000 neuro-monitored thyroidectomies series compared the surgical outcomes between the EBD group (Ligasure) and conventional group, including 23 patients with Graves’ disease. The EBD group had overall lower surgical complication rates in comparison with the conventional group. Effective hemostasis helps to avoid excessive muscle retraction and reduce the injury of the extralaryngeal muscles, thereby avoiding the influence of the extralaryngeal muscles on the fine control of voice production after thyroid surgery. In the current study, applying EBDs in modern thyroid surgery for Graves’ disease brought a better objective voice outcome in PR and Jitter. The continuous developments of novel nerve monitoring and hemostasis devices can further change the surgical procedures; in addition, the anti-adhesive material/technique also show great potential for improving surgical outcomes.

PH is one of the most common complications after TT [18], particularly in Graves’ disease [19,20,21]. Graves’ disease patients usually have a large thyroid volume and a vigorous vessel supply. Dissecting the thyroid lobe in a narrow space is susceptible to bleeding and increases the risk of PG damage due to poor visualization, especially in procedures performed by low-volume surgeons [22]. In a study of 42 Graves’ disease patients, Nair et al. [20] reported temporary and permanent hypocalcemia in 42.85% and 9.52% of patients, respectively. Similarly, in another study of 165 Graves’ disease patients, Guo et al. [21] reported temporary and permanent hypocalcemia in 18.8% and 3.6% of patients, respectively. In the current study, Group A patients treated by the surgeon in his first decade of practice had high rates of temporary and permanent PH (39.1% and 8.7%, respectively). In Group B, the improved outcomes on the temporary and permanent PH rates (18.2% and 0.0%, respectively) may be attributable to improvements in surgical technique and strategies for preserving PG function. In Group A patients, we preferred the strategy of at least one PG autotransplantation, which in the 1990s, was believed to have a low risk of permanent hypoparathyroidism [23,24]. Our current principles of intraoperative PG management is in situ preservation, and the procedures include (1) division of individual blood vessels near the thyroid gland to avoid interference with the blood supply to PGs, (2) careful inspection of the thyroid capsule to determine whether PGs had adhered to the capsule, (3) routine check of PG blood supply with stabbing test (stabbing PG with a 23 G needle to check if fresh blood flowed out) or nick test (incision made on PG capsule with fine scissors to check if fresh blood flowed out) when a disturbance of blood supply was suspected, and (4) PG autotransplantation only in the case of devascularization [13].

Weiss et al. reported a 1.34% postoperative hematoma rate and a 0.32% mortality rate in 150,012 thyroid surgery patients [25]. The thyroid gland is a highly vascularized organ [26], particularly in Graves’ disease. Hematoma is reported in a large proportion of Graves’ disease patients who undergo thyroid surgeries [27]. In the current study, Group A had a higher hematoma rate compared with Group B, but the difference was not statistically significant (10.5% vs. 2.3%, *p* = 0.108). Some studies have reported that the use of EBDs decreases the occurrence of hematoma after thyroidectomy [28,29]. However, others have shown that hematoma does not significantly differ between EBDs and the conventional clamp-and-tie technique [30,31,32]. Therefore, the advantage of EBDs needs further study in a high volume of Graves’ disease patients.

Surgeon performance can be evaluated by the occurrence of postoperative complications, and the experience of surgeons can be roughly estimated by their age or their years of surgical practice. For young surgeons, the importance of education and training to gain experience is obvious [33]. To achieve the best results, surgeons need time to acquire the necessary technical background and to master routine procedures [34,35]. Experts typically reach their peak performance after approximately 10 years of experience in their specialty [36]. However, the surgical performance of older surgeons might decline over time due to mental fatigue from performing repetitive procedures, physiological factors, lack of updated knowledge, and poor adherence to principles of evidence-based medicine and new techniques, all of which can contribute to reduced safety and poor treatment outcomes [14,37].

According to some studies, thyroid surgeries performed by experienced surgeons tend to have low complication rates, short lengths of hospital stay [38,39], and low costs of treating complications [39]. However, defining an experienced thyroid surgeon is difficult and requires the consideration of factors other than age, years of practice, and number of thyroidectomy procedures performed [33]. For example, Duclos et al. demonstrated high rates of complications after thyroidectomy in procedures performed by inexperienced surgeons and older surgeons [14]. Generally, an experienced thyroid surgeon can be defined as a surgeon who has (1) has at least 10 years of experience in thyroid surgery, (2) an average annual volume of 100 thyroidectomies with a low complication rate [40], and (3) familiarity with new technologies, e.g., IONM and EBDs. This study showed that, as surgeon experience increases, the rate of major postoperative complication rates in TT for Graves’ disease decreases.

Talent and experience are insufficient to ensure a safe surgery if a surgeon lacks the motivation and willingness to progress [41]. To maintain a high level of performance for the rest of their careers, surgeons must continuously evaluate the quality of the care they deliver and update their surgical concepts. As the result in this study, we suggest that the inexperienced surgeons should improve the technique to safely preserve the function of RLN and PGs in regular thyroid surgeries before performing operations on Graves’ disease patients.

This study had some limitations. As this was not a prospective or randomized study, bias resulting from comparisons of patients treated in different periods and by different techniques and instruments was unavoidable. However, the data were collected from a single surgeon over his 30 years of experience in performing surgery in patients with Graves’ disease. This study was indeed comparing the different experience levels with surgical performance from the same surgeon. The description about “peak performance” in this article is the general situation of the surgeon’s career. However, in the development of thyroid surgery, the progress in IONM and EBDs in the past decade was too remarkable, which had a very positive impact on the surgical outcomes. Therefore, accumulation of surgical experience and the utilization of newly developed devices are the indispensable factors for thyroid surgeons in this era to improve patient safety. This was also the specific reason for choosing the two time periods (Group A and Group B) for comparison.

## 5. Conclusions

TT for Graves’ disease is a challenging procedure with a high rate of major complications. To reduce the major complications rate, inexperienced thyroid surgeons should remain vigilant when performing total thyroidectomy for Graves’ disease. It is also suggested that inexperienced surgeons should improve the technique to safely preserve the function of RLN and PGs in regular thyroid surgeries before performing operations on Graves’ disease patients. Updates on surgical concepts and the effective use of operative adjuncts (i.e., EBD) are necessary to improve patient safety and functional outcomes.

## Figures and Tables

**Table 1 jcm-11-01298-t001:** Demographic and clinical characteristics of Group A and Group B.

	Group A (46 Patients)	Group B (44 Patients)	*p* Value
Gender			0.697
Women	35	35	
Men	11	9	
Age (Year, Mean ± SD)	35.8 ± 13.8	40.8 ± 14.1	0.096
Pathology			0.489
Benign (%)	41 (89.1)	37 (84.1)	
Malignancy (%)	5 (10.9)	7 (15.9)	
PG autotransplantation * (%)	40 (87.0)	22 (50.0)	0.001

* The number of patients with at least one parathyroid gland (PG) autotransplantation.

**Table 2 jcm-11-01298-t002:** Major complications of total thyroidectomy in Graves’ Disease.

	Group A (46 Patients)	Group B (44 Patients)	*p* Value
RLN palsy ^a,b^ (%)	12/92 (13.0)	1/88 (1.1)	0.001
Temporary (%)	11 (11.9)	1 (1.1)	
Permanent (%)	1 (1.1)	0 (0.0)	
Postoperative hypocalcemia (%)	22/46 (47.8)	8/44 (18.2)	0.002
Temporary (%)	18 (39.1)	8 (18.2)	
Permanent (%)	4 (8.7)	0 (0.0)	
Postoperative hematoma (%)	5/46 (10.9)	1/44 (2.3)	0.108
Thyroid storm (%)	1/46 (2.2)	0/44 (0.0)	0.323

^a^ Recurrent laryngeal nerve (RLN) palsy was considered permanent if vocal cord dysfunction persisted longer than 6 months after surgery. The incidence was based on the number of RLNs at risk. ^b^ No occurrence of bilateral RLN palsy.

**Table 3 jcm-11-01298-t003:** Six-week postoperative objective voice analysis in Group B with or without EBD.

	Without EBD (22 Patients)	With EBD (22 Patients)	*p* Value
Fmin decrease > 30%	2 (9.1)	3 (13.6)	0.635
Fmax decrease > 30%	10 (45.5)	5 (22.7)	0.112
PR decrease > 30%	9 (40.9)	2 (9.1)	0.015 *
Mean F0 decrease > 30%	2 (9.1)	2 (9.1)	1.000
Jitter increase > 30%	14 (63.6)	7 (31.8)	0.035 *
Shimmer increase > 30%	6 (27.3)	5 (22.7)	0.728
NHR increase > 30%	5 (22.7)	3 (13.6)	0.434

Abbreviation: EBD = Energy-based devices; Fmin = Minimum frequency; Fmax = Maximum frequency; PR = Pitch range; Mean F0 = Mean fundamental frequency; NHR = Noise-harmonic ratio. * *p* < 0.05 was considered statistically significant.

## Data Availability

The original contributions presented in the study are included in the article. Further inquiries can be directed to the corresponding authors.

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
