# Peer review of "Accumulation of Experience and Newly Developed Devices Can Improve the Safety and Voice Outcome of Total Thyroidectomy for Graves’ Disease"

_jcm, 2022, doi:10.3390/jcm11051298_

Round 1

Reviewer 1 Report

In the manuscript "Accumulation of experience and newly developed devices can improve the safety and voice outcome of total thyroidectomy for Graves’ disease," the authors performed a retrospective study on the surgical experience of patients with Graves disease and the surgical outcomes. The paper certainly has clinical significance. However, some minor criticisms are present, as follows:

- A more detailed discussion on the patients with malignancy and if they had any re-interventions and/or iodine treatment and the total outcome of these cases.

A more detailed outcome of the patient who presented thyroid storm after surgery should be exciting, and the postoperative management and vocal effects.

- What were the outcomes on the extreme cases, and what should be done if malignancy is discovered? Does this affect the voice outcome?

- Please incorporate this idea in the conclusions.

Author Response

Author's Response

Dear Reviewer,

We deeply appreciate your comments.

We have revised our manuscript in-line with the comment made.

The followings are our response:

Response to the Reviewer #1

In the manuscript "Accumulation of experience and newly developed devices can improve the safety and voice outcome of total thyroidectomy for Graves’ disease," the authors performed a retrospective study on the surgical experience of patients with Graves disease and the surgical outcomes. The paper certainly has clinical significance. However, some minor criticisms are present, as follows:

Comment-1:

A more detailed discussion on the patients with malignancy and if they had any re-interventions and/or iodine treatment and the total outcome of these cases.

A more detailed outcome of the patient who presented thyroid storm after surgery should be exciting, and the postoperative management and vocal effects.

What were the outcomes on the extreme cases, and what should be done if malignancy is discovered? Does this affect the voice outcome?

Please incorporate this idea in the conclusions.

Response:

Thank you for the comment. We provided our data in the result section as “Among the patients in Group B, 3 of 7 patients had malignant pathology result received radioactive iodine treatment, and none of the 7 patients received re-intervention or had cancer recurrence.

The postoperative voice outcome of Graves’ disease complicated with thyroid cancer are indeed worthy of further research, but they are limited to a small number of cases to obtain meaningful results in current study. In the future, specific studies can be designed and conducted as multi academic center research. Thank you for your precious suggestion.

In addition, the time of radioactive iodine treatment in our institution was always more than 6 weeks after surgery. The discussion on the effect of radioactive iodine on voice is an interesting topic, which is not limited to patients with Graves’ disease, and deserves a special study to survey.

In this study, there was only one case of thyroid storm in group A, and there were no routine clinical measures about voice function at that time. As you mentioned, the assessment of thyroid storm with today's follow-up assessments must provide a lot of valuable information, and we are also interested. Thank you for your precious comment.

We thank you for your valued comments and suggestions, which we feel substantially improve our manuscript, and hope that the revisions meet with your approval.

Sincerely,

Feng-Yu Chiang, M.D.,

Department of Otolaryngology-Head and Neck Surgery, E-Da Hospital, I-Shou University, Kaohsiung, Taiwan

Address: No. 1, Yida Road, Jiaosu Village, Yanchao District, Kaohsiung 824, Taiwan.

E-mails: fychiang@kmu.edu.tw

Cheng-Hsun Chuang, M.D., Chi-Feng Tai, M.D.

Department of Otorhinolaryngology–Head and Neck Surgery, Kaohsiung Medical University Hospital, Kaohsiung Medical University, Kaohsiung, Taiwan.

Address: 100TzYou 1st Road, Kaohsiung 807, Taiwan.

E-mails: 18chengxun@gmail.com (C-H.C.); cftai@kmu.edu.tw (C-F.T.)

 (On behalf of all coauthors)

Feb 24, 2022

Reviewer 2 Report

Congratulation for the author and surgeon for his 30 years of experience in thyroid surgery. As well as for the idea of the work, which I find very interesting. However the results seems to be quite obvious, that experience of the the surgeon and new devices will improve the outcomes. 

Specific comments: 

line 236:

"Experts typically reach their peak performance after approximately 10 years of experience in their specialty [36]. However, the surgical performance of older surgeons might decline over  time due to mental fatigue from performing repetitive procedures, physiological factors, lack of updated knowledge, and poor adherence to principles of evidence-based medicine and new techniques,..."

My question is why you compare 1th and 3 th decade (1990-1999 -Group A), and (2010-2019 -Group B) if the peak performance of your surgeon should be in 2000-2010? 

line 102- intraoperative neuromonitoring (IONM), - what type of neuromonitoring was used? 

single thyroid surgeon (F.-Y.C.) within a 30-year period - it is not indicated what was his annual volume and complication rates ?

line 176 "Although the RLN was visually identified in all cases in Group A, 11 176 temporary palsies" visual identification does not prevent palsies especailly those temporary. 

line 193:  "Effective hemostasis helps to avoid excessive muscle retraction, reduces the production of postoperative, which affects the fine extralaryngeal muscle control of voice."  something is missing in this sentence - I do not understand it?

line 198: "in addition, anti-adhesive material/technique and minimal invasive procedure (i.e. thyroid ablation procedures) also show great potential for improving surgical outcomes.- what type of anti adhesive material? technique? - ablation procedures - what kind of indication in Graves Besedov? and thyroid ablation is no longer a surgery.

line 219:  routine check of PG blood supply with stabbing test or nick test when disturbance of blood supply was suspected" can you explain what kind of tests they are?

Author Response

Author's Response

Dear Reviewer,

We deeply appreciate your comments.

We have revised our manuscript in-line with the comment made.

The followings are our response:

Response to the Reviewer #2

Congratulation for the author and surgeon for his 30 years of experience in thyroid surgery. As well as for the idea of the work, which I find very interesting. However the results seems to be quite obvious, that experience of the the surgeon and new devices will improve the outcomes.

Comment-1:

line 236:

"Experts typically reach their peak performance after approximately 10 years of experience in their specialty [36]. However, the surgical performance of older surgeons might decline over time due to mental fatigue from performing repetitive procedures, physiological factors, lack of updated knowledge, and poor adherence to principles of evidence-based medicine and new techniques,..."

My question is why you compare 1th and 3 th decade (1990-1999 -Group A), and (2010-2019 -Group B) if the peak performance of your surgeon should be in 2000-2010?

Response:

Thanks for your very important comment. The description about “peak performance” in this article is the general situation of the surgeon's career. However, in the development of thyroid surgery, the progress in neuromonitoring and energy-based device in the past decade was too remarkable, which had a very positive impact on the surgical outcomes. At the same time, the accumulation of surgical experience is still continuing, fortunately, there has not been a decline in my surgical skills. There are the specific reasons for choosing these two periods of time to compare.

Comment-2:

    line 102- intraoperative neuromonitoring (IONM), - what type of neuromonitoring was used?

Response:

        Thank you for your comment. We added “intermittent intraoperative neuromonitoring (IONM)” in the description.

Comment-3:

    single thyroid surgeon (F.-Y.C.) within a 30-year period - it is not indicated what was his annual volume and complication rates?

Response:

Thank you for your comment, we added the description in Materials and Methods section as “The surgeon performs more than 200 thyroid surgeries every year, and the surgical com-plications in recent years were presented in the reference [12,13].

Comment-4:

    line 176 "Although the RLN was visually identified in all cases in Group A, 11 temporary palsies" visual identification does not prevent palsies especailly those temporary.
    line 193:  "Effective hemostasis helps to avoid excessive muscle retraction, reduces the production of postoperative, which affects the fine extralaryngeal muscle control of voice."  something is missing in this sentence - I do not understand it?

Response:

        Thank you for your suggestion. We added the description (Line 176) as “In other words, the visual integrity of the RLN is not consistent with the functional preservation, especially the temporary nerve injuries.”; and replaced the description (Line 193) as “Effective hemostasis helps to avoid excessive muscle retraction and reduce the injury of the extralaryngeal muscles, thereby avoiding the influence of the extralaryngeal muscles on the fine control of voice production after thyroid surgery.” The original sentence did have an input error, thank you for your correction.

Comment-5:

    line 198: "in addition, anti-adhesive material/technique and minimal invasive procedure (i.e. thyroid ablation procedures) also show great potential for improving surgical outcomes.- what type of anti adhesive material? technique? - ablation procedures - what kind of indication in Graves Besedov? and thyroid ablation is no longer a surgery.

Response:

        To avoid confusion for readers, we have removed the description about thyroid ablation. Thank you for your comment.

Comment-7:

line 219: routine check of PG blood supply with stabbing test or nick test when disturbance of blood supply was suspected" can you explain what kind of tests they are?

Response:

        The description about the stabbing test (stabbing PG with a 23G needle to check if fresh blood flowed out) and nick test (incision mad on PG capsule with fine scissors to check if fresh blood flowed out) were from the reference 13 [Comparison of hypocalcemia rates between LigaSure and clamp‐and‐tie hemostatic technique in total thyroidectomies. Head & neck 2019, 41, 3677-3683]. We have modified the description as your suggestion, thank you for your comment.

We thank you for your valued comments and suggestions, which we feel substantially improve our manuscript, and hope that the revisions meet with your approval.

Sincerely,

Feng-Yu Chiang, M.D.,

Department of Otolaryngology-Head and Neck Surgery, E-Da Hospital, I-Shou University, Kaohsiung, Taiwan

Address: No. 1, Yida Road, Jiaosu Village, Yanchao District, Kaohsiung 824, Taiwan.

E-mails: fychiang@kmu.edu.tw

Cheng-Hsun Chuang, M.D., Chi-Feng Tai, M.D.

Department of Otorhinolaryngology–Head and Neck Surgery, Kaohsiung Medical University Hospital, Kaohsiung Medical University, Kaohsiung, Taiwan.

Address: 100TzYou 1st Road, Kaohsiung 807, Taiwan.

E-mails: 18chengxun@gmail.com (C-H.C.); cftai@kmu.edu.tw (C-F.T.)

 (On behalf of all coauthors)

Feb 24, 2022

Round 2

Reviewer 2 Report

Thank you for respons to my questions. 

The main doubts explained:

"Thanks for your very important comment. The description about “peak performance” in this article is the general situation of the surgeon's career. However, in the development of thyroid surgery, the progress in neuromonitoring and energy-based device in the past decade was too remarkable, which had a very positive impact on the surgical outcomes. At the same time, the accumulation of surgical experience is still continuing, fortunately, there has not been a decline in my surgical skills. There are the specific reasons for choosing these two periods of time to compare."

Now it is clear why you choose those periods, however I still believe that your work would be more interesting by adding this 3th period, as you have reached peak performance but progress in new technique was not that clear. At least pleas add this explanation in the manuscript. 

Author Response

Author's Response

Dear Reviewer and Editor,

We deeply appreciate your comments.

We have revised our manuscript in-line with the comment made.

The followings are our response:

Response to the Reviewer #2 (Round 2)

Comment-1:

The main doubts explained:

"Thanks for your very important comment. The description about “peak performance” in this article is the general situation of the surgeon's career. However, in the development of thyroid surgery, the progress in neuromonitoring and energy-based device in the past decade was too remarkable, which had a very positive impact on the surgical outcomes. At the same time, the accumulation of surgical experience is still continuing, fortunately, there has not been a decline in my surgical skills. There are the specific reasons for choosing these two periods of time to compare."

Now it is clear why you choose those periods, however I still believe that your work would be more interesting by adding this 3th period, as you have reached peak performance but progress in new technique was not that clear. At least pleas add this explanation in the manuscript.

Response:

        We added the description in Discussion/Limitation section as “The description about “peak performance” in this article is the general situation of the surgeon's career. However, in the development of thyroid surgery, the progress in IONM and EBD in the past decade was too remarkable, which had a very positive impact on the surgical outcomes. Therefore, accumulation of surgical experience and utilization of newly developed devices are the indispensable factors for thyroid surgeons in this era to improve patient safety. This was also the specific reason for choosing the two time periods (Group A and Group B) for comparison.” We thank you for your precious comments and suggestions, which we feel substantially improve our manuscript.

Sincerely,

Feng-Yu Chiang, M.D.,

Department of Otolaryngology-Head and Neck Surgery, E-Da Hospital, I-Shou University, Kaohsiung, Taiwan

Address: No. 1, Yida Road, Jiaosu Village, Yanchao District, Kaohsiung 824, Taiwan.

E-mails: fychiang@kmu.edu.tw

Cheng-Hsun Chuang, M.D., Chi-Feng Tai, M.D.

Department of Otorhinolaryngology–Head and Neck Surgery, Kaohsiung Medical University Hospital, Kaohsiung Medical University, Kaohsiung, Taiwan.

Address: 100TzYou 1st Road, Kaohsiung 807, Taiwan.

E-mails: 18chengxun@gmail.com (C-H.C.); cftai@kmu.edu.tw (C-F.T.)

 (On behalf of all coauthors)

Feb 24, 2022
